# Symmetries in Weight Space Learning: To Retain or Remove?

## Abstract

Weight-space learning, an emerging paradigm that studies neural networks through their parameter space, has shown promise for tasks ranging from predicting model behavior to addressing privacy risks. An important caveat in weight-space learning is that neural networks admit extensive *parameter symmetries*: distinct weight configurations can implement the same function. Such symmetries have been studied from multiple angles and play an important role in both theory and practice, including Low-Rank Adaptation (LoRA), a state-of-the-art fine-tuning method for large language models (LLMs) that exhibits scale and rotational invariances. In this paper, we present a *theoretical* study of symmetries in weight-space learning and ask: What is the appropriate problem formulation in the presence of symmetries (e.g., those induced by LoRA), and should redundant representations that encode the same end-to-end function be removed? We answer this by showing that whether redundancy matters depends on the target functional of interest. In particular, we prove that end-to-end symmetries (such as those in LoRA) should *not* always be quotiented out: doing so can compromise universality for classes of weight-space prediction tasks. To our knowledge, this is the first formal identification of this phenomenon, offering principled guidance for the design of weight-space methods across many applications.

## 1 Introduction

Weight space learning refers to the task of using a model's parameters to predict properties that are implicitly encoded in its weight space. This problem has practical applications in areas such as privacy leakage, sensitivity analysis, generalization prediction, and model behavior forecasting. With the rise of LLMs and the abundance of publicly available fine-tuned models, weight space learning has recently garnered significant interest within the deep learning community.

Low-Rank Adaptation (LoRA) is a state-of-the-art fine-tuning method for Large Language Models (LLMs). It aims to reduce the computational cost of full-parameter fine-tuning by learning low-rank updates to the model's weights. The primary objective is to efficiently adapt a pretrained model to new data while ensuring the updates remain meaningful with respect to the fine-tuning dataset. Therefore, the LoRA weight space encodes partial information about the fine-tuning data, as expected, since the method is explicitly designed to learn from it. However, this property raises important privacy concerns, as fine-tuning datasets often contain sensitive information. Beyond privacy, various characteristics of the fine-tuned model, such as its sensitivity to weight perturbations, its generalization ability, and its behavior on specific data subsets, are correlated with, and can potentially be inferred from, the information embedded in the weight space.

Independently, almost every real-world neural network classes exhibit symmetries in their weight spaces, that is, different sets of parameters can produce the same end-to-end function. For example, in LoRA, the low-rank factors can be scaled or rotated without altering the resulting function. Similar symmetries appear in other neural architectures, such as neuron permutation invariance in feedforward networks and scaling invariance in ReLU-based models and equivariant networks.

In this paper, we conduct a *theoretical* study of the problem of weight space learning, with a particular focus on understanding the role of *parameter symmetries* in this setting. A natural question arises in this investigation: Should we remove these symmetries and use invariant neural networks (i.e.,

quotiening out the symmetries) to process the weight space in a more efficient was (i.e., exploiting symmetries), or should we retain the original weights in their raw form?

As a first step toward addressing this foundational question, we work with a general formulation of weight space learning, and demonstrate a perhaps surprising result:

> Removing symmetries can, in some cases, compromise the expressive power (or universality) of the weight space learning problem, even when the model itself exhibits symmetries.

Specifically, we show that the symmetries relevant to weight space learning are *only* a subset of the symmetries of the underlying model. In some cases, the weight space learning problem exhibits no symmetries at all, even when the original model is symmetric. In other words, while all the function class is symmetric to a set of transformation of weights, there are instances of weight space learning that are non-invariant to all such transformations. Consequently, we conclude that the decision to remove or retain symmetries in weight space learning depends on the structure of the downstream task, as there is no universal rule that guarantees expressivity preservation across all settings.

This result lays a conceptual foundation for handling symmetries in weight-space learning in future work. In our main results, we explicitly construct symmetry-free instances and analyze how they relate to the symmetries present in LoRA weight space.

In short, in this paper we make the following contributions:

- We initiate a *theoretical* study of symmetries in weight space learning, which is a new paradign aiming to learn features of neural networks from their weight (i.e., parameter) space. This is closely related to *meta-learning* task, and it has shown to be an emerging.

- We present the first general analysis of symmetries in weight space learning with a particular results: even if the function space possess symmetries, removing symmerries and invariant leraning for weight space learning will compromise the expressive power. The result holds generally, and the decision to either remve sy,metreis or not should be task specific and need model or task evaluations.

## 2 PROBLEM STATEMENT

Consider a model $f(x; \mathbf{w})$, where $x \in \mathcal{X}$ denotes the input and $\mathbf{w} \in \mathcal{W}$ represents the learnable parameters (weights). Both $\mathcal{X}$ and $\mathcal{W}$ are assumed to be complete metric spaces. The associated function space is defined as

$$\mathcal{F} := \{f(\cdot; \mathbf{w}) \mid \mathbf{w} \in \mathcal{W}\}.$$

The goal of the *weight space learning* problem is the following: given a dataset of function-label pairs $(f(\cdot; \mathbf{w}_i), y_i) \in \mathcal{F} \times \mathbb{R}$ for $i \in [n]$, the task is to learn a meta-regression function $\hat{f}_{\text{meta}} : \mathcal{W} \to \mathbb{R}$ that not only predicts $y_i$ accurately on observed weights $\mathbf{w}_i$, but also generalizes well to unseen weights $\mathbf{w} \in \mathcal{W}$.

In practice, different parameter values can correspond to the same function. That is, the mapping $\mathbf{w} \mapsto f(\cdot; \mathbf{w}) \in \mathcal{F}$ is generally not injective. A notable example arises in the LoRA formulation for fine-tuning neural networks. Let $\mathbf{W} \in \mathbb{R}^{d \times d}$ denote the model weights. In LoRA, the weights are parameterized as

$$\mathbf{W} = \mathbf{W}_0 + A^\top B, \quad \text{with } A, B \in \mathbb{R}^{r \times p},$$

where $\mathbf{W}_0 \in \mathbb{R}^{p \times p}$ denotes frozen pre-trained weights, and $A, B$ are low-rank matrices learned during fine-tuning, with $r \ll p$.

In this formulation, any invertible matrix $C \in \mathbb{R}^{r \times r}$ induces an equivalence relation, since

$$\mathbf{W} = \mathbf{W}_0 + (C^{-1}A)^\top C^\top B = \mathbf{W}_0 + A^\top B.$$

Thus, the weight-space representations $(A, B)$ and $(C^{-1}A, C^\top B)$ define the same function.

Weight space symmetries are not unique to LoRA. Other notable examples include permutation symmetries among neurons in feedforward networks, and scaling symmetries in ReLU networks, where appropriate rescaling of adjacent layers can leave the output function unchanged.

In this paper, we formalize a *general framework* to study weight space learning under such symmetries. Specifically, we consider a group $G$ that acts continuously on the weight space $\mathcal{W}$, where $\mathcal{W} \subseteq \mathbb{R}^d$ is a domain of parameters. We assume that $G$ fully captures the symmetry structure of the model in the sense that

$$f(\cdot, \mathbf{w}) \equiv f(\cdot, g\mathbf{w}) \quad \forall g \in G, \, \mathbf{w} \in \mathcal{W},$$

where $g\mathbf{w}$ denotes the group action of $g \in G$ on the parameter $\mathbf{w} \in \mathcal{W}$.

## 3 MAIN RESULTS

In this section we present our main results. We begin with a warm-up on zeroth-order weight-space learning, showing that there exists a class of weight-space functionals that inherits all parameter symmetries present in the model. We then consider a natural generalization, higher-order weight-space learning, and show that these symmetries can break; we specifically analyze what happens in the LoRA weight space in this regime. Finally, we provide theoretical results demonstrating the existence of symmetry-free weight-space tasks even when the underlying model admits rich parameter symmetries.

### 3.1 ZEROTH-ORDER WEIGHT SPACE LEARNING

Consider a weight space learning problem with data $(f(\cdot; \mathbf{w}_i), y_i) \in \mathcal{F} \times \mathbb{R}$ for $i \in [n]$, where the goal is to learn a meta-regression function of the form:

$$\mathsf{F}(\mathbf{w}; \phi, \psi) := \phi \left( \int_{\mathcal{X}} \psi\left(f(x; \mathbf{w})\right) d\mu(x) \right), \tag{1}$$

where $\phi$ and $\psi$ are parametrized functions (to be learned from the dataset), and $\mathcal{X}$ is a measurable subset of $\mathbb{R}^d$. In practice, this corresponds to learning zeroth-order features from the model $f(x; \mathbf{w})$, using samples from the input domain $\mathcal{X}$ to approximate the integral and train $\phi$ and $\psi$.

In this setting, we can directly observe that:

$$\mathsf{F}(g\mathbf{w}; \phi, \psi) = \phi \left( \int_{\mathcal{X}} \psi\left(f(x; g\mathbf{w})\right) dx \right) \tag{2}$$

$$= \phi \left( \int_{\mathcal{X}} \psi\left(f(x; \mathbf{w})\right) dx \right) \tag{3}$$

$$= \mathsf{F}(\mathbf{w}; \phi, \psi), \tag{4}$$

since $f(x; g\mathbf{w}) = f(x; \mathbf{w})$ for all $g \in G$. This implies that zeroth-order features in weight space learning are invariant under the parameter symmetries of the function space. Consequently, standard methods for learning under symmetries can be effectively applied in such cases.

In particular, we obtain the following result for LoRA weights:

**Corollary 3.1.** *Weight space learning with LoRA is* $\mathrm{GL}_r(\mathbb{R})$*-invariant for zeroth-order meta-regression functions (Equation* (1)*). This symmetry matches that of the original LoRA formulation.*

Note that $\mathrm{GL}_r(\mathbb{R})$ denotes the set of all invertible matrices $C \in \mathbb{R}^{r \times r}$.

### 3.2 HIGHER-ORDER WEIGHT SPACE LEARNING

In this subsection, we consider meta-regression functions that require features beyond zeroth order. As a concrete example, consider weight space learning on the zero-loss manifold of a pre-trained neural network, with training data $(x_j, z_j) \in \mathcal{X} \times \mathbb{R}, j \in [J]$. Under the square loss, the optimization objective is as follows:

$$L(\mathbf{w}) := \frac{1}{2J} \sum_{j=1}^{J} (f(x_j; \mathbf{w}) - z_j)^2, \quad s.t. \quad \mathbf{w} \in \mathbb{R}^d. \tag{5}$$

Now, note that we have

$$L(\mathbf{w}) = \frac{1}{2J} \sum_{j=1}^{J} \big(f(x_j; \mathbf{w}) - z_j\big)^2, \qquad r_j(\mathbf{w}) := f(x_j; \mathbf{w}) - z_j, \quad \forall j \in [J], \tag{6}$$

$$\implies \nabla_{\mathbf{w}} L(\mathbf{w}) = \frac{1}{J} \sum_{j=1}^{J} r_j(\mathbf{w}) \, \nabla_{\mathbf{w}} f(x_j; \mathbf{w}), \tag{7}$$

$$\implies \nabla_{\mathbf{w}}^2 L(\mathbf{w}) = \frac{1}{J} \sum_{j=1}^{J} \Big[ \nabla_{\mathbf{w}} f(x_j; \mathbf{w}) \, \nabla_{\mathbf{w}} f(x_j; \mathbf{w})^\top + r_j(\mathbf{w}) \, \nabla_{\mathbf{w}}^2 f(x_j; \mathbf{w}) \Big]. \tag{8}$$

Around the zero-loss manifold (i.e., after pretraining), we have $r_j(\mathbf{w}) = 0$ for all $j \in [J]$; in other words, the data are already interpolated. However, in neural network optimization this is usually not sufficient for generalization. Indeed, reaching zero training loss is often straightforward. One must continue training beyond interpolation, which amounts to optimizing *on* the zero-loss manifold; see, for example, Sharpness-Aware Minimization (SAM) (Foret et al., 2020).

Under this condition, we conclude that

$$\nabla_{\mathbf{w}}^2 L(\mathbf{w}) = \frac{1}{J} \sum_{j=1}^{J} \nabla_{\mathbf{w}} f(x_j; \mathbf{w}) \, \nabla_{\mathbf{w}} f(x_j; \mathbf{w})^\top. \tag{9}$$

In particular, consider the function

$$\mathsf{F}(\mathbf{w}) := \mathrm{Tr}\left(\nabla_{\mathbf{w}}^2 L(\mathbf{w})\right) = \frac{1}{J} \sum_{j=1}^{J} \nabla_{\mathbf{w}} \, \mathrm{Tr}\left(f(x_j; \mathbf{w}) \, \nabla_{\mathbf{w}} f(x_j; \mathbf{w})^\top\right) \tag{10}$$

$$= \frac{1}{J} \sum_{j=1}^{J} \nabla_{\mathbf{w}} \, \mathrm{Tr}\left(\nabla_{\mathbf{w}} f(x_j; \mathbf{w})^\top \nabla_{\mathbf{w}} f(x_j; \mathbf{w})\right) \tag{11}$$

$$= \frac{1}{J} \sum_{j=1}^{J} \|\nabla_{\mathbf{w}} f(x_j; \mathbf{w})\|_2^2, \tag{12}$$

which captures the average *sensitivity* of the model $f(\cdot; \mathbf{w})$ with respect to its parameters. This quantity is essential in the study of neural network weight quantization, where the goal is to map trained weights to a quantized set with minimal error.

Can a zeroth-order meta-regression function learn such sensitivity features? Let us examine the symmetries of the *sensitivity function* $\mathsf{F}(\mathbf{w})$ in the LoRA weight space, where $\mathbf{w} = (A, B)$. In this setting, for any $g \in G$ corresponding to $C \in \mathrm{GL}_r(\mathbb{R})$, as we discussed, the two configurations $(C^{-1}A, C^\top B)$ and $(A, B)$ are functionally equivalent. However, the sensitivity transforms as

$$\mathsf{F}(g\mathbf{w}) = \frac{1}{J} \sum_{j=1}^{J} \sum_{k=1}^{p} \left\{ \|C^\top \nabla_{\mathbf{A}_k} f(x_j; \mathbf{w})\|_2^2 + \|C^{-1} \nabla_{\mathbf{B}_k} f(x_j; \mathbf{w})\|_2^2 \right\}, \tag{13}$$

where $\mathbf{A}_k$ denotes the $k$-th column of matrix $A$ (similarly for $B$). This follows straightforwardly by computing the gradients with respect to the parameters $\mathbf{w} := (A, B)$. Note that in this example, the group of symmetries is $G \equiv \mathrm{GL}_r(\mathbb{R})$ with elements $g \equiv C \in G$.

From the above expression, we have $\mathsf{F}(\mathbf{w}) = \mathsf{F}(g\mathbf{w})$, for all $g$ if and only if $C \in \mathrm{O}(r)$. Indeed, if $C = c I_{r \times r}$ for some scaler $c \neq 1$, then we have (generically) $\mathsf{F}(\mathbf{w}) \neq \mathsf{F}(g\mathbf{w})$.

In other words, while all invertible matrices $C \in \mathrm{GL}_r(\mathbb{R})$ correspond to weight space symmetries of LoRA, only *orthogonal matrices* $C \in \mathrm{O}(r)$, defined as $\mathrm{O}(r) := \{C^\top C = I_{r \times r}\}$, preserve the sensitivity function (since they preserve $\ell_2$-norms of gradients involved in the above formula). Thus, learning sensitivity-based features requires going beyond zeroth-order meta-regressors.

**Corollary 3.2.** *Weight-space learning with LoRA is only $\mathrm{O}(r)$-invariant when learning sensitivity-dependent features (Equation (12)). This invariance group is a strict subset of the full symmetry group of the original LoRA formulation, which is $\mathrm{GL}_r(\mathbb{R})$.*

*Remark* 3.3. The above illustrates that the symmetries relevant for weight-space learning can be a *strict* subset of the symmetries of the underlying model. Furthermore, compressing the weight space based solely on the model's full symmetry group can compromise universality (expressive power), as certain weight-space learning tasks (such as those involving sensitivity) require reduced symmetry and retain full representations of parameters.

### 3.3 WEIGHT-SPACE LEARNING WITH NO SYMMETRY

Consider the following meta-regressor:

$$\mathsf{F}(\mathbf{w}) := \partial_1[L(\mathbf{w})], \tag{14}$$

where $L(\mathbf{w}) = \frac{1}{2J}\sum_{j=1}^{J}(f(x_j; \mathbf{w}) - z_j)^2$ is as previously defined, and $\partial_1[\cdot]$ denotes the partial derivative with respect to the first coordinate of the vector $\mathbf{w} \in \mathbb{R}^d$. Here, we do not assume interpolation, and thus $r_j(\mathbf{w}) = f(x_j; \mathbf{w}) - z_j$ are not necessarily zero, $j \in [J]$.

In this case, what symmetries, if any, does $\mathsf{F}(\mathbf{w})$ exhibit? Assume that $\rho(g) \in \mathrm{GL}_d(\mathbb{R})$ denotes the matrix representation of a group element $g \in G$, such that the group action is given by $g\mathbf{w} := \rho(g)\mathbf{w}$. Then, we have

$$\mathsf{F}(g\mathbf{w}) = \mathsf{F}(\mathbf{w}) \quad \text{only if} \quad \rho(g) = \begin{pmatrix} 1 & 0 & \cdots & 0 \\ 0 & * & \cdots & * \\ \vdots & \vdots & \ddots & \vdots \\ 0 & * & \cdots & * \end{pmatrix}. \tag{15}$$

To see how this happens, note that

$$\mathsf{F}(\mathbf{w}) = \frac{1}{2J}\sum_{j=1}^{J}\partial_1[(f(x_j; \mathbf{w}) - z_j)^2] \tag{16}$$

$$= \frac{1}{J}\sum_{j=1}^{J}f(x_j; \mathbf{w}) - z_j)\partial_1[f(x_j; \mathbf{w})]. \tag{17}$$

Therefore

$$\mathsf{F}(g\mathbf{w}) = \frac{1}{J}\sum_{j=1}^{J}f(x_j; g\mathbf{w}) - z_j)\left(\partial_1[f(x_j; \mathbf{w})]\right)_{\mathbf{w}=g\mathbf{w}} \tag{18}$$

$$= \frac{1}{J}\sum_{j=1}^{J}f(x_j; g\mathbf{w}) - z_j)\left(\partial_1[f(x_j; \mathbf{w})]\right)_{\mathbf{w}=g\mathbf{w}}, \tag{19}$$

where $\left(\partial_1[f(x_j; \mathbf{w})]\right)_{\mathbf{w}=g\mathbf{w}}$ means the gradient of the function with respect to its first coordinate, computed for $g\mathbf{w}$ instead of $\mathbf{w}$. The above functional identity can only hold (generically) if and only if we have $\partial_1[f(x; \mathbf{w})]$ for all $x$, when considered as a function of $\mathbf{w} \in \mathbb{R}^d$, is invariance under transformation $\mathbf{w} \mapsto g\mathbf{w}$, for all $g \in G$. For generic $f$, this can only happen if (at least) $\rho(g)\mathbf{w}$ and $\mathbf{w}$ agree on their first coordinate for all $g \in G$ and $\mathbf{w}$. This means that $\rho(g)$ has to be act trivially on the first coordinate of the parameters.

This result holds analogously for other indices $i = 2, 3, \ldots, d$. This means that, there exists no universal group of symmetries of the general tasks in weight-space learning, since the intersection of all the above conditions for $i \in [d]$ results having $\rho(g) = I_{d \times d}$, which trivializes the set of possible symmetries in the parameter space. Indeed, even when using first-order derivatives of the square-loss as features, the only allowable transformation is identity, implying that no nontrivial symmetries are preserved, despite the underlying model potentially being symmetric. This means that:

> The general problem of weight-space learning (beyond the zeroth-order case) requires considering the full weight space under its inherent symmetries. In other words, any compression of the weight space that eliminates these symmetries compromises the universality (i.e., expressive power) of the weight-space learning framework.

However, the above result holds only in the context of the general formulation. For restricted weight-space learning meta-regressors (such as the zeroth-order function class), it may be possible to remove certain symmetries from the model's weight space while still maintaining full expressivity. Identifying an appropriate space for weight-space learning thus heavily depends on the downstream task, and no universal solution exists.

### 3.4 SYMMETRIES IN HIGHER-ORDER FEATURES

The explanation presented in the previous subsections extends beyond the square loss and applies to a broader class of functionals than the trace of the Hessian. We investigate this class of functionals and their associated symmetries here.

Fix a positive integer $k \in \mathbb{N}$, which we call the order of the weight-space features. Let $\mathbf{w} \in \mathbb{R}^d$ denote the weights, and suppose a group $G$ acts on the weight space via invertible matrices:

$$g\mathbf{w} := \rho(g)\,\mathbf{w}, \quad \rho(g) \in \mathbb{R}^{d \times d}.$$

Moreover, recall that $f(x; \mathbf{w}) \equiv f\big(x; g\mathbf{w}\big)$ for all $g \in G$.

Under these conditions, define the *order-$k$ features* as follows.

**Definition 3.4.** The *order-$k$ features* functionals $\mathsf{F} : \mathbb{R}^d \to \mathbb{R}$ given (for each $x$) by

$$\mathsf{F}(\mathbf{w}; x) := \big\langle a,\ \nabla_{\mathbf{w}}^{\otimes k} f(x; \mathbf{w}) \big\rangle \in \mathbb{R}, \tag{20}$$

where $\nabla_{\mathbf{w}}^{\otimes k} f(x; \mathbf{w}) \in (\mathbb{R}^d)^{\otimes k}$ is the order-$k$ tensor of all partial derivatives of $f(x; \mathbf{w})$ with respect to $\mathbf{w}$, and $a \in (\mathbb{R}^d)^{\otimes k}$ is a fixed coefficient tensor.

For technical reasons, without loss of generality, we assume $a \in \mathrm{Sym}^k(\mathbb{R}^d)$ which means it is a symmetric tensor. The above features encode all information obtainable by differentiating with respect to the weights up to order $k$. For instance, the previously defined sensitivity, the trace of the Hessian, is a special case with $k = 2$.

How can we leverage order-$k$ features to define functionals on the weight space? There is a canonical construction, motivated by two-layer neural networks, which yields the following order-$k$ weight-space functionals:

$$\mathsf{F}(\mathbf{w}) := \phi\left( \int_{\mathcal{X}} \psi(\mathsf{F}(\mathbf{w}; x))\, \mathrm{d}\mu(x) \right), \tag{21}$$

where $\phi$ and $\psi$ are continuous functions, and $\mu$ is a measure. Note that if $\mathsf{F}(\mathbf{w}) = \mathsf{F}(g\mathbf{w})$ for generic $\phi, \psi, \mu$, then by choosing Dirac measures $\mu = \delta_x$ and taking the identity functions for $\phi$ and $\psi$, we conclude that $\mathsf{F}(g\mathbf{w}; x) = \mathsf{F}(\mathbf{w}; x)$. Therefore, to understand the symmetries of order-$k$ weight-space functionals, it suffices to study the symmetries of the order-$k$ features.

**Question:** *What is the group of symmetries of $\mathsf{F}(\mathbf{w}; x)$ for generic $f$ and $x$? In other words, can we have $\mathsf{F}(g\mathbf{w}; x) = \mathsf{F}(\mathbf{w}; x)$ for all $x$, $f$, and all $g \in G$?*

We provide an answer to the above question, which serves as a theoretical foundation for higher-order features in weight-space learning.

**Theorem 3.5.** *We have $\mathsf{F}(g\mathbf{w}; x) = \mathsf{F}(\mathbf{w}; x)$ for all $x$, $f$, and $g \in G$ if and only if*

$$\rho(g)^{\otimes k} a = a, \quad \forall\, g \in G. \tag{22}$$

This result is important because it provides a concrete criterion for testing whether higher-order features are invariant. For instance, to double-check the trace of the Hessian ($k = 2$), take $a = \mathrm{vec}(I_{d \times d})$, the vectorization of the identity. We then require $\rho(g)^{\otimes 2} a = a$ for all $g \in G$.

Using properties of tensor products,

$$(\rho(g) \otimes \rho(g))\, a = (\rho(g) \otimes \rho(g))\, \mathrm{vec}(I_{d \times d})$$
$$= \mathrm{vec}\big(\rho(g)\, I_{d \times d}\, \rho(g)^{\mathsf{T}}\big)$$
$$= \mathrm{vec}\big(\rho(g)\rho(g)^{\mathsf{T}}\big).$$

Therefore, if $(\rho(g) \otimes \rho(g))\, a = a$ for $a = \text{vec}(I_{d \times d})$, we necessarily have $\rho(g)\rho(g)^{\mathsf{T}} = I_{d \times d}$. In other words, $\rho(g) \in O(d)$ is an orthogonal transformation. This condition shows that the result we presented for LoRA extends to *any* linear group of symmetries: for the trace of the Hessian, the only allowed symmetry is orthogonal transformations. Any other group element (such as scalings) violates the condition and is thus symmetry-breaking. Indeed, we also have a converse result.

**Proposition 3.6.** *Assume*

$$\rho(g)^{\otimes k} a = a, \quad \forall \rho(g) \in O(d). \tag{23}$$

*In other words, the weight-space functional is invariant to the entire orthogonal group. Then one necessarily has*

$$a = \text{vec}\big(I_d^{\otimes k}\big). \tag{24}$$

*This means that the only orthogonally symmetric weight-space functionals are traces of higher-order tensors.*

**Corollary 3.7.** *The group of symmetries of the order-$k$ feature*

$$\mathsf{F}(\mathbf{w}; x) := \big\langle a, \nabla_{\mathbf{w}}^{\otimes k} f(x; \mathbf{w}) \big\rangle \in \mathbb{R},$$

*for fixed $a \in (\mathbb{R}^d)^{\otimes k}$, is the* isotropy *group of $a$:*

$$G(a) := \big\{ g \in G : \rho(g)^{\otimes k} a = a \big\} \subseteq G. \tag{25}$$

*In particular, in many cases $G(a) \subsetneq G$.*

A natural question arises: how often do we have $G(a) = G$? Is the strict inclusion $G(a) \subsetneq G$ pathological or typical? The following theorem addresses this.

**Theorem 3.8.** *Assume the group $G$ acts on the weight space* faithfully, *meaning that for any nontrivial $g \in G$, the map $\mathbf{w} \mapsto g\mathbf{w}$ is not the identity. Assume further that either $k$ is odd, or there exists $g \in G$ such that $\rho(g) \notin \{I_{d \times d}, -I_{d \times d}\}$. Then $G(a) = \{\mathsf{e}\}$ for almost every $a \in (\mathbb{R}^d)^{\otimes k}$ (with respect to Lebesgue measure), where $\mathsf{e}$ denotes the identity element of $G$.*

*In other words, order-$k$ features $\mathsf{F}(\mathbf{w}; x)$, and the order-$k$ weight-space functionals $\mathsf{F}(\mathbf{w})$ derived from them, have* no *symmetries, even when the model admits extensive parameter symmetries.*

This result is significant: avoiding pathological cases, for almost any coefficient tensor $a$, the induced weight-space functionals of order $k$ exhibit no symmetries. Symmetries in weight-space learning should therefore be considered with this caveat. Blindly quotienting out parameter symmetries can compromise universality, since there are tasks with no such symmetries that would become unattainable.

**Theorem 3.9.** *Define the subspace*

$$A_G := \Big\{ a \in (\mathbb{R}^d)^{\otimes k} : \rho(g)^{\otimes k} a = a \ \forall g \in G \Big\}. \tag{26}$$

*In words, $A_G$ consists of all order-$k$ fully symmetric features. Suppose the representation has a spectral gap in the sense that there exists a constant[1] $c < 1$, independent of $d$, such that for each nontrivial $g \in G$,*

$$\big| \text{Tr}(\rho(g)) \big| \le c\, d.$$

*Then, while $\dim\big((\mathbb{R}^d)^{\otimes k}\big) = d^k$, we have*

$$\dim(A_G) \le c\, d^k.$$

## 4 EXPERIMENTS

As a proof of concept experiment, here we consider the sensitivity of a LoRA fine-tuned test loss with respect to its LoRA factors. We show that without modifying the end-to-end function, the sensitivity changes. This demonstrates that parameter symmetries for zeroth-order properties do not necessarily hold for gradient-based properties.

---

[1]Such $c < 1$ always exist provided that a finite group acts faithfully; see Remark 10.2.

| Model \ $\sigma$ | $10^{-7}$ | $10^{-6}$ | $10^{-5}$ | $10^{-4}$ | $10^{-3}$ | $10^{-2}$ | $10^{-1}$ | $3\cdot 10^{-1}$ | $5\cdot 10^{-4}$ |
|---|---|---|---|---|---|---|---|---|---|
| Baseline | **0.9634** | **0.9634** | **0.9634** | **0.9634** | **0.9634** | 0.9636 | 0.9917 | 1.9998 | 4.7824 |
| Balanced $(P_i^*)^2$ | **0.9634** | **0.9634** | **0.9634** | **0.9634** | **0.9634** | **0.9635** | **0.9898** | **1.3452** | **1.7136** |
| Scaled $r = 10^2$ | **0.9634** | **0.9634** | **0.9634** | 0.9649 | 1.1836 | 16.4109 | 20.4792 | 20.2002 | 17.9131 |
| Scaled $r = 10^3$ | **0.9634** | **0.9634** | 0.9641 | 1.5015 | 20.1965 | 20.2504 | 23.8895 | 21.9865 | 18.0028 |
| Scaled $r = 10^4$ | **0.9634** | 0.9642 | 1.0733 | 14.3455 | 18.1590 | 18.7772 | 24.4439 | 22.4494 | 18.1642 |
| Scaled $r = 10^5$ | 0.9639 | 1.1025 | 13.8518 | 18.9335 | 22.0456 | 18.1350 | 24.3826 | 22.4639 | 18.1619 |
| Unperturbed | **0.9634** | | | | | | | | |

Table 1: Averaged test loss under perturbations with standard deviation $\sigma$ across different LoRA factorizations of the same model. Lower is better.

Given a model with $n$ LoRA layers $A_i, B_i \in \mathbb{R}^{r,d}$ and $i \in [n]$, we rescale the LoRA factors with a factor $r \in \mathbb{R}^+$ such that

$$\tilde{A}_i = \frac{1}{r} A_i \text{ and } \tilde{B}_i = r B_i.$$

Additionally, we also consider the factorization that balances the sum of the squares of the Frobenius norm of the LoRA factors of each layer. For a single layer, this factorization solves

$$\min_{P_i \in GL_n(\mathbb{R})} \left\| P A_i \right\|_F^2 + \left\| P^{-T} B_i \right\|_F^2.$$

Solving this optimization problem yields

$$P_i^* = ((AA^T)^{-\frac{1}{2}}((AA^T)^{\frac{1}{2}}(BB^T)(AA^T)^{\frac{1}{2}})^{\frac{1}{2}}(AA^T)^{-\frac{1}{2}})^{\frac{1}{2}}.$$

Applying the transformation $(P_i^*)^2$ empirically proves to lower the sensitivity of the loss with respect to weight perturbations.

$$\tilde{A}_i = (P_i^*)^2 A_i \text{ and } \tilde{B}_i = (P_i^*)^{-2} B_i.$$

We define the model's LoRA sensitivity as the expected change in loss for a normally distributed perturbation on its LoRA weights

$$f_{\text{sens}}(\mathbf{w}) = \mathbb{E}_{\epsilon \sim \mathcal{N}(0,\sigma^2)}[L(\mathbf{w} + \epsilon) - L(w + \epsilon)].$$

This weight space function is higher-order since it is not only based on the end-to-end model evaluation but is based on the response in output values to changes in parameters. In this experiment, we show that this sensitivity weight space function does not respect the parameter symmetries of the model. That is, we consider actions on the weight space $g \in \text{GL}_r(\mathbb{R})$ under which the model is invariant $f(\cdot, \mathbf{w}) \equiv f(\cdot, g\mathbf{w}) \quad \forall g \in G, \mathbf{w} \in \mathcal{W}$. We then show, that these actions do in fact change the sensitivity weight space function

$$f_{\text{sens}}(g\mathbf{w}) \neq f_{\text{sens}}(\mathbf{w}).$$

As a baseline, we fine-tune Llama 3.2 3B using LoRA on the GSM8K dataset and compare its loss under weight perturbations to different factorizations of the same LoRA model. In particular, we consider linear rescaling with a range of magnitudes. For all three model factorizations – baseline, rescale, and balanced – we analyze the effect of LoRA weight perturbations. We draw 5 different perturbations $\epsilon_i \sim \mathcal{N}(0, \sigma_i^2)$ for each standard deviation considered $\sigma_i$. We report the test loss of each perturbed model as well as the loss of the unperturbed model. Since we are using invertible transformations to refactor the LoRA parameters, all unperturbed models share the same test loss.

Table 11 shows that different perturbations are indeed significantly prone to performance reductions under weight perturbations. These results show that even though a model might exhibit symmetries to a set of transformations, there generally exist weight space functions that break these symmetries. This experiment also shows that our proposed transformation $P^*$ can even improve weight perturbation robustness beyond the baseline.

## 5 CONCLUSION

We presented a theoretical study of the weight-space learning problem, with a focus on a general formulation. Neural networks typically exhibit extensive parameter symmetries, meaning different weights can encode the same function. A common instinct is therefore to quotient out such transformations and perform invariant learning to improve robustness and sample efficiency. It is natural to apply the same idea to weight-space learning. However, our results reveal a surprising twist: for classes of functions with parameter symmetries, blindly quotienting out symmetries can compromise universality—there exist tasks that become unsolvable once these symmetries are removed. We provide several theoretical results, including instances relevant to LoRA, showing that the choice to remove or retain symmetries depends critically on the downstream objective; there is no one-size-fits-all rule. Taken together, our findings offer a conceptual foundation and a practical guideline: treat symmetry handling as a task-dependent design choice when learning from neural network weights, rather than a universal preprocessing step.

## 6 RELATED WORK

It has long been known that neural network parameter spaces possess symmetries (Brea et al., 2019; Ainsworth et al., 2022; Zhou et al., 2023; Meng et al., 2018; Martens & Grosse, 2015; Xie & Smidt, 2025; Ziyin et al., 2025; da Silva et al., 2025), and exploiting these has led to advances across optimization and model merging. Moreover, geometric understanding of the loss landscape has been an active research topic; examples include exploiting weight-space geometry to avoid sharp minima (Foret et al., 2020) and using the NTK (Jacot et al., 2018) to explain aspects of neural network training. Other related directions include the lottery ticket hypothesis (Frankle et al., 2020) and studies of mode connectivity and loss surfaces (Garipov et al., 2018; Li et al., 2018; Mehta et al., 2021).

Another active area is learning to optimize, a meta-learning approach (Finn et al., 2017; Nichol et al., 2018; Li et al., 2017; Rusu et al., 2018; Rajeswaran et al., 2019) that designs optimizers to generalize across tasks (Thérien et al., 2024; Shen et al., 2020; Metz et al., 2021; Gao et al., 2022; Flennerhag et al., 2023). Recent work investigates how symmetry in parameter space can improve generalization in this setting (Zamir et al., 2025; Zhao et al., 2022; 2023; Yang et al., 2023). For more on learned gradient methods and hypernetworks, see Andrychowicz et al. (2016); Ha et al. (2016).

Low-Rank Adaptation (LoRA) accelerates the fine-tuning of large language models by representing weight updates as low-rank matrices, thereby reducing trainable parameters (Hu et al., 2021). Since its introduction, many extensions have appeared, including LoRA concatenation for skill composition (Prabhakar et al., 2024), QLoRA for quantized fine-tuning (Dettmers et al., 2024), alternative initialization strategies such as PiSSA (Meng et al., 2024), and theoretical investigations into LoRA's expressive power (Zeng & Lee, 2023) (see also (Bałazy et al., 2024)). To bridge the performance gap between LoRA and full fine-tuning, techniques include fixed learning-rate adjustments for the LoRA factors (Hayou et al., 2024), reformulating the gradient update with a low-rank structure (Wang et al., 2024), and scale-invariant optimization strategies (Yen et al., 2024; Li et al., 2024).

Learning on LoRA (LoL) (Putterman et al., 2024) uses trained LoRA weights as inputs to a meta-network for downstream prediction tasks; the meta-network can infer dataset properties (e.g., size) or fine-tuned model characteristics (e.g., accuracy). In such setups, handling the symmetries of LoRA weights is crucial for robust generalization. From an applications perspective, Salama et al. (2024) show how LoRA weights can estimate training-set size. More ambitiously, Haim et al. (2022) demonstrate that image-level training data can be recovered from the weights of a fully connected network. In contrast, Elbaz et al. (2024) find that for group-invariant networks, such reconstructions often converge to different but functionally equivalent samples, and propose a workaround using task-specific priors. Finally, Horwitz et al. (2024) propose a mixture-of-experts approach that organizes fine-tuned models into a hierarchical structure based on foundation-model lineage, enabling weight-space reasoning. A related direction is probing in weight space: Kahana et al. (2024) propose learning structured probes via latent factorization, offering a principled way to extract interpretable signals from weights.

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

## 7 PROOF OF THEOREM 3.5

We restate and prove the desired result as follows. Throughout the proof, $G$ acts on the weight space via a representation $\rho : G \to \mathrm{GL}_d(\mathbb{R})$, denoted by $g\mathbf{w} = \rho(g)\mathbf{w}$, and we use the standard Frobenius inner product on tensors. As in the setup, any $f$ is invariant under the group action, i.e.,

$$f(x; \rho(g)\mathbf{w}) = f(x; \mathbf{w}) \qquad \forall\, g \in G,\ x,\ \mathbf{w}. \tag{27}$$

Given $a \in (\mathbb{R}^d)^{\otimes k}$, recall the order-$k$ feature

$$\mathsf{F}(\mathbf{w}; x) := \big\langle a,\ \nabla_{\mathbf{w}}^{\otimes k} f(x; \mathbf{w}) \big\rangle.$$

**Theorem 7.1.** *We have $\mathsf{F}(\rho(g)\mathbf{w}; x) = \mathsf{F}(\mathbf{w}; x)$ for all $x$, all $\mathbf{w}$, all $g \in G$, and all functions $f$ satisfying Equation* (27)*, if and only if*

$$\rho(g)^{\otimes k} a = a, \qquad \forall\, g \in G. \tag{28}$$

*Proof.* We first review transformation rules for higher derivatives under the (linear) change of variables $\mathbf{u} = \rho(g)\mathbf{w}$. Differentiating Equation (27) with respect to $\mathbf{w}$ and applying the chain rule yields, for each $k \geq 1$,

$$\nabla_{\mathbf{w}}^{\otimes k} f(x; \rho(g)\mathbf{w}) = \big(\rho(g)^{-\mathsf{T}}\big)^{\otimes k} \nabla_{\mathbf{w}}^{\otimes k} f(x; \mathbf{w}). \tag{29}$$

(For intuition, consider $k = 1$ which gives us $\nabla_{\mathbf{w}} f(x; \rho(g)\mathbf{w}) = \rho(g)^{-\mathsf{T}} \nabla_{\mathbf{w}} f(x; \mathbf{w})$; higher orders follow by iterating the chain rule.)

*($\Rightarrow$)* Suppose $\mathsf{F}(\rho(g)\mathbf{w}; x) = \mathsf{F}(\mathbf{w}; x)$ for all $x, \mathbf{w}$ and all $f$ satisfying Equation (27). Using Equation (29) and the Frobenius pairing,

$$\mathsf{F}(\rho(g)\mathbf{w}; x) = \Big\langle a,\ \nabla_{\mathbf{w}}^{\otimes k} f(x; \rho(g)\mathbf{w}) \Big\rangle = \Big\langle a,\ \big(\rho(g)^{-\mathsf{T}}\big)^{\otimes k} \nabla_{\mathbf{w}}^{\otimes k} f(x; \mathbf{w}) \Big\rangle$$
$$= \Big\langle \rho(g)^{\otimes k} a,\ \nabla_{\mathbf{w}}^{\otimes k} f(x; \mathbf{w}) \Big\rangle,$$

where in the last step we used $\langle a, (M^{-\mathsf{T}})^{\otimes k} T \rangle = \langle M^{\otimes k} a, T \rangle$ for any $M$ and tensor $T$ of matching order. By the assumed symmetry, we have

$$\Big\langle \rho(g)^{\otimes k} a - a,\ \nabla_{\mathbf{w}}^{\otimes k} f(x; \mathbf{w}) \Big\rangle = 0 \quad \text{for all } x, \mathbf{w},\ \text{and all } f \text{ obeying Equation (27)}.$$

Varying $f$ and $\mathbf{w}$, the set of attainable $k$th derivative tensors $\nabla_{\mathbf{w}}^{\otimes k} f(x; \mathbf{w})$ spans the space of symmetric order-$k$ tensors. Hence the linear functional induced by $\rho(g)^{\otimes k} a - a$ vanishes on the set of all symmetric tensors, denoted by $\mathrm{Sym}^k(\mathbb{R}^d)$, which forces $\rho(g)^{\otimes k} a = a$ and the equality holds for all $g \in G$. This is exactly Equation (28).

*($\Leftarrow$)* Conversely, assume Equation (28). Then, using Equation (29),

$$\mathsf{F}(\rho(g)\mathbf{w}; x) = \Big\langle a,\ \big(\rho(g)^{-\mathsf{T}}\big)^{\otimes k} \nabla_{\mathbf{w}}^{\otimes k} f(x; \mathbf{w}) \Big\rangle = \Big\langle \rho(g)^{\otimes k} a,\ \nabla_{\mathbf{w}}^{\otimes k} f(x; \mathbf{w}) \Big\rangle$$
$$= \Big\langle a,\ \nabla_{\mathbf{w}}^{\otimes k} f(x; \mathbf{w}) \Big\rangle = \mathsf{F}(\mathbf{w}; x),$$

for all $x, \mathbf{w}$ and all $f$ satisfying Equation (27). This proves the claimed invariance. $\square$

*Remark* 7.2. Since the $k$th derivative tensor of a scalar function is symmetric, only the symmetric component of $a$ contributes to $\mathsf{F}(\mathbf{w}; x)$. The condition Equation (28) should therefore be understood on the (relevant) symmetric subspace $\mathrm{Sym}^k(\mathbb{R}^d)$; the statement above is written in the ambient tensor space for notational simplicity.

## 8 PROOF OF PROPOSITION 3.6

We prove that orthogonal symmetries forces the coefficient tensor to be (constant factors of) the canonical "trace" tensor.

**Proposition 8.1.** *Assume*

$$\rho(g)^{\otimes k} a \;=\; a, \qquad \forall\, \rho(g) \in O(d). \tag{30}$$

*In other words, the weight-space feature $\mathsf{F}(\mathbf{w}; x) = \langle a, \nabla_{\mathbf{w}}^{\otimes k} f(x; \mathbf{w}) \rangle$ is symmetric under the whole orthogonal group. Then, one necessarily has*

$$a \;=\; \mathrm{vec}\big(I_d^{\otimes k}\big), \tag{31}$$

*i.e., the only orthogonally symmetric weight-space functionals are (normalized) traces of higher-order tensors.*

*Proof.* For a scalar-valued $f$, the tensor $\nabla_{\mathbf{w}}^{\otimes k} f(x; \mathbf{w})$ is symmetric in its $k$ indices. Hence only the fully symmetrized part of $a$ contributes to the pairing:

$$\big\langle a, \nabla_{\mathbf{w}}^{\otimes k} f \big\rangle \;=\; \big\langle \mathrm{Sym}_k(a), \nabla_{\mathbf{w}}^{\otimes k} f \big\rangle,$$

where $\mathrm{Sym}_k$ denotes the average over all index permutations. Moreover, Equation (30) implies $\rho(g)^{\otimes k} \mathrm{Sym}_k(a) = \mathrm{Sym}_k(a)$ for all $g$ because $\mathrm{Sym}_k$ commutes with $\rho(g)^{\otimes k}$. Thus, without loss of generality, we may assume $a \in \mathrm{Sym}^k(\mathbb{R}^d)$ (fully symmetric).

Take $g = -I_d \in O(d)$. From Equation (30),

$$(-I_d)^{\otimes k} a \;=\; a \;\implies\; (-1)^k a \;=\; a.$$

Hence, if $k$ is odd then $a = 0$ (the feature is identically zero). The only nontrivial case is even $k = 2m$; we assume this onwards.

Associate to $a \in \mathrm{Sym}^{2m}(\mathbb{R}^d)$ the homogeneous polynomial of degree $2m$

$$p_a(u) \;:=\; \big\langle a, u^{\otimes 2m} \big\rangle, \qquad u \in \mathbb{R}^d.$$

Orthogonal symmetries of $a$ is equivalent to $p_a(gu) = p_a(u)$ for all $g \in O(d)$:

$$p_a(gu) = \big\langle a, (gu)^{\otimes 2m} \big\rangle = \big\langle (\rho(g)^{\otimes 2m})^\mathsf{T} a, u^{\otimes 2m} \big\rangle = \big\langle a, u^{\otimes 2m} \big\rangle = p_a(u).$$

Thus $p_a$ is an $O(d)$-invariant homogeneous polynomial of degree $2m$.

Note that the orthogonal group acts transitively on each sphere $\{u : \|u\|_2 = r\}$. Therefore, $p_a$ must be constant on spheres, i.e., $p_a(u) = q(\|u\|_2)$ for some univariate function $q$. Since $p_a$ is homogeneous of degree $2m$, it follows that

$$p_a(u) \;=\; c\,\|u\|_2^{2m} \;=\; c\,(u^\top u)^m$$

for some constant $c \in \mathbb{R}$.

Let

$$J \;:=\; \sum_{i_1,\ldots,i_m=1}^{d} \big(e_{i_1} \otimes e_{i_1}\big) \otimes \cdots \otimes \big(e_{i_m} \otimes e_{i_m}\big) \;=\; \Big( \sum_{i=1}^{d} e_i \otimes e_i \Big)^{\otimes m} \;\in\; \mathrm{Sym}^{2m}(\mathbb{R}^d),$$

where $\{e_i\}$ is the standard basis. A direct check shows that for all $u$,

$$\big\langle J, u^{\otimes 2m} \big\rangle = \Big( \sum_{i=1}^{d} u_i^2 \Big)^m = \|u\|_2^{2m}.$$

Hence $p_J(u) = \|u\|_2^{2m}$, and thus we must have $p_a(u) = c\, p_J(u)$ for all $u$. By the injectivity of the polarization map $a \mapsto p_a$ on $\mathrm{Sym}^{2m}(\mathbb{R}^d)$, it follows that

$$a \;=\; c\, J.$$

Note, the tensor $J$ is exactly the complete symmetrization of the $m$-fold Kronecker power of the metric (identity) and corresponds, under the flattening convention used in the main text, to $\mathrm{vec}(I_d^{\otimes 2m})$. Thus

$$a \;=\; c\, \mathrm{vec}\big(I_d^{\otimes k}\big), \qquad k = 2m.$$

Since an overall scalar does not affect the symmetry property (it just rescales the feature), we may normalize $a$ so that $c = 1$, yielding the stated form. $\qquad\square$

*Remark* 8.2. For odd $k$ the only $O(d)$-invariant coefficient is $a = 0$, so nontrivial orthogonally invariant features exist only for even order. Within $\mathrm{Sym}^k(\mathbb{R}^d)$ and for even $k$, the $O(d)$-invariant subspace is one-dimensional, generated by the repeated metric; this is the precise sense in which the only orthogonally symmetric weight-space functionals are (normalized) traces of higher-order tensors.

# 9 PROOF OF THEOREM 9.1

We prove that, under a faithful weight-space action, a generic coefficient tensor has trivial isotropy for the $k$-fold tensor representation.

**Theorem 9.1.** *Assume the group $G$ acts on the weight space faithfully, i.e., for any nontrivial $g \in G$, the map $\mathbf{w} \mapsto g\mathbf{w} = \rho(g)\mathbf{w}$ is not the identity. Assume further that either $k$ is odd, or there exists $g \in G$ such that $\rho(g) \notin \{I_{d \times d}, -I_{d \times d}\}$. Then*

$$G(a) = \{\mathsf{e}\} \quad \text{for almost every } a \in (\mathbb{R}^d)^{\otimes k}$$

*with respect to Lebesgue measure, where $\mathsf{e}$ denotes the identity element of $G$. Equivalently, for almost every $a$, the only symmetry of the order-$k$ feature $\mathsf{F}(\mathbf{w}; x) = \langle a, \nabla_{\mathbf{w}}^{\otimes k} f(x; \mathbf{w}) \rangle$ is the identity, even if the model admits extensive parameter symmetries.*

*Proof.* Write $V := \mathbb{R}^d$ and $W := V^{\otimes k}$. For each $g \in G$, let

$$\mathrm{Fix}(g) := \{ a \in W : \rho(g)^{\otimes k} a = a \} = \ker\big(\rho(g)^{\otimes k} - I_W\big)$$

be the fixed-point subspace of $\rho(g)^{\otimes k}$; this is a linear (hence measurable) subspace of $W$.

We claim that for every nontrivial $g \in G$, the subspace $\mathrm{Fix}(g)$ is proper (i.e., $\mathrm{Fix}(g) \neq W$), except in the degenerate case $\rho(g) = -I_{d \times d}$ with even $k$.

Indeed, if $\rho(g) = I_{d \times d}$ then $g$ is trivial by faithfulness, so we exclude this case. If $\rho(g) = -I_{d \times d}$ and $k$ is even, then

$$\rho(g)^{\otimes k} = (-I_{d \times d})^{\otimes k} = I_W,$$

hence $\mathrm{Fix}(g) = W$. In all remaining cases ($\rho(g) \neq I_{d \times d}$ and either $k$ is odd, or $\rho(g) \notin \{\pm I_{d \times d}\}$), we show $\rho(g)^{\otimes k} \neq I_W$, which implies $\mathrm{Fix}(g) \subsetneq W$.

To see this, complexify and put $M := \rho(g)$. Over $\mathbb{C}$, $M$ is triangularizable, so $M = SJS^{-1}$ with $J$ upper triangular and not a scalar multiple of the identity (since $M \neq \pm I$ in the even-$k$ branch, and $M \neq I$ in the odd-$k$ branch). Then

$$M^{\otimes k} = (S^{\otimes k})(J^{\otimes k})\big(S^{-1}\big)^{\otimes k}.$$

If $J$ is not a scalar matrix, then $J^{\otimes k}$ is not a scalar matrix either, hence $M^{\otimes k} \neq I_W$. If $J = \lambda I$ is scalar, then $M = \lambda I$ with $\lambda \neq 1$; over $\mathbb{R}$ the only such scalar orthogonal possibilities are $\lambda = -1$. In the odd-$k$ branch, $(-I)^{\otimes k} = -I_W \neq I_W$; in the even-$k$ branch, $\lambda = -1$ is exactly the excluded degenerate case. Thus, outside the excluded case, $M^{\otimes k} \neq I_W$, proving $\mathrm{Fix}(g) \subsetneq W$.

Assume first that $G$ is finite or countable (or its action is continuous). For any nontrivial $g$, $\mathrm{Fix}(g)$ is a proper linear subspace of $W$, hence of Lebesgue measure zero. Therefore, the set

$$\mathcal{S} := \bigcup_{g \in G \setminus \{\mathsf{e}\}} \mathrm{Fix}(g)$$

is a countable union of measure-zero sets and thus has measure zero. By definition, $a \notin \mathcal{S}$ if and only if $\rho(g)^{\otimes k} a \neq a$ for all nontrivial $g$, i.e., $G(a) = \{\mathsf{e}\}$. This proves the claim for finite or countable $G$.

For compact Lie group $G$ (or, more generally, for closed subgroups of $\mathrm{GL}(d, \mathbb{R})$) with faithful $\rho$ and under the same exclusion of degeneracy, one can use to the principal orbit type theorem: there exists an open dense (hence full-measure) subset of $W$ on which all stabilizers are conjugate and of minimal dimension. Because $\ker(\rho^{\otimes k}) = \{\mathsf{e}\}$ under the present hypotheses, the minimal stabilizer is trivial, yielding the same conclusion.

Combining these cases establishes the theorem. $\qquad\square$

*Remark* 9.2. If $k$ is even and there exists $g$ with $\rho(g) = -I_{d \times d}$, then $\rho(g)^{\otimes k} = I$ and *every* $a$ is fixed by $g$; in that case the conclusion fails. The theorem is therefore meaningful precisely when this degeneracy is absent (which is guaranteed when $k$ is odd, or when no nontrivial element acts as $-I$).

For finite or countable $G$, the proof reduces to a measure-zero union of proper fixed-point subspaces. For compact Lie groups, the same generic triviality follows from standard orbit-type stratification results.

## 10 PROOF OF THEOREM 3.9

**Theorem 10.1.** *Define*

$$A_G := \Big\{ a \in (\mathbb{R}^d)^{\otimes k} : \rho(g)^{\otimes k} a = a \ \forall g \in G \Big\}.$$

*Suppose there exists a constant $c < 1$, independent of d, such that for each nontrivial $g \in G$,*

$$\big| \operatorname{Tr}(\rho(g)) \big| \leq c\, d.$$

*Then, since* $\dim\big((\mathbb{R}^d)^{\otimes k}\big) = d^k$, *we have*

$$\dim(A_G) \ \leq \ c\, d^k.$$

*Proof.* Let $\mu$ be a probability measure on $G$ which is bi-invariant under group multiplication (the counting probability measure if $G$ is finite; Haar probability measure if $G$ is compact). Consider the averaging operator (Reynolds operator)

$$P \ := \ \int_G \rho(g)^{\otimes k}\, \mathrm{d}\mu(g) \ : \ (\mathbb{R}^d)^{\otimes k} \to (\mathbb{R}^d)^{\otimes k}.$$

For every $h \in G$,

$$\rho(h)^{\otimes k} P = \int_G \rho(hg)^{\otimes k}\, \mathrm{d}\mu(g) = \int_G \rho(g)^{\otimes k}\, \mathrm{d}\mu(g) = P,$$

where bi-invariance of $\mu$ is used in the middle equality. Hence $\operatorname{Im}(P) \subseteq A_G$. Conversely, if $a \in A_G$ then $\rho(g)^{\otimes k} a = a$ for all $g$, so $Pa = a$. Therefore $P$ is the projector onto $A_G$ and $\dim(A_G) = \operatorname{rank}(P) = \operatorname{Tr}(P)$.

By linearity of trace and the identity $\operatorname{Tr}(A \otimes B) = \operatorname{Tr}(A)\operatorname{Tr}(B)$, we have

$$\dim(A_G) = \operatorname{Tr}(P) = \int_G \operatorname{Tr}\big(\rho(g)^{\otimes k}\big)\, \mathrm{d}\mu(g) = \int_G \big(\operatorname{Tr} \rho(g)\big)^k\, \mathrm{d}\mu(g).$$

Since $\dim(A_G) \geq 0$, we can bound it by the integral of absolute values:

$$\dim(A_G) \ \leq \ \int_G \big|\operatorname{Tr} \rho(g)\big|^k\, \mathrm{d}\mu(g).$$

If $G$ is finite with $|G| < \infty$ and $\mu$ is the uniform measure,

$$\dim(A_G) = \frac{1}{|G|}\Big( d^k + \sum_{g \neq \mathsf{e}} \big(\operatorname{Tr} \rho(g)\big)^k \Big) \ \leq \ \frac{1}{|G|} d^k + \frac{|G|-1}{|G|}\, c^k\, d^k \ \leq \ \Big(\frac{1}{|G|} + c\Big) d^k.$$

In particular, whenever $|G| \geq 1/c$ this yields $\dim(A_G) \leq c\, d^k$. (If $|G| < 1/c$, the bound becomes $\dim(A_G) \leq \max\{c, 1/|G|\}\, d^k$, which still shows a strict sublinear fraction of the ambient dimension because $c < 1$ and $1/|G| < 1$.) For infinite groups, the proof follows similarly.

Combining the two cases completes the proof. $\qquad\square$

*Remark* 10.2. Note that the assumption $c < 1$ is always satisfied for faithful finite group actions. To see this, let us briefly recall that a group $G$ acts faithfully on a (real) vector space if and only if its matrix representation $\rho(g) \in \mathbb{R}^{d \times d}$ satisfies $\rho(g) \neq I_{d \times d}$ whenever $g \neq \operatorname{id}_G$, the identity element of the group. Moreover, the matrices $\rho(g)$, $g \in G$, are all orthogonal, which implies that all eigenvalues of $\rho(g)$ lie on the unit circle in the complex plane. Therefore, $|\operatorname{Tr}(\rho(g))|$ is at most $d$, and equality holds only if all eigenvalues are equal to 1, which (for orthogonal matrices) implies $\rho(g) = I_{d \times d}$. This is a contradiction unless $g$ is the identity. Hence, for all nontrivial $g \in G$ one must have $|\operatorname{Tr}(\rho(g))| < d$, which means that a constant $c < 1$ exists satisfying the assumption in the theorem.

| Model $\diagdown$ $\sigma$ | Unpert. | $10^{-7}$ | $10^{-6}$ | $10^{-5}$ | $10^{-4}$ | $10^{-3}$ | $10^{-2}$ | $10^{-1}$ | $3\cdot10^{-1}$ | $5\cdot10^{-1}$ |
|---|---|---|---|---|---|---|---|---|---|---|
| Baseline | | **3.2777** | **3.2777** | **3.2777** | **3.2777** | **3.2777** | **3.2787** | 3.4709 | 8.3331 | 9.8147 |
| Balanced $(P_i^*)^2$ | | **3.2777** | **3.2777** | **3.2777** | **3.2777** | **3.2777** | **3.2787** | **3.3746** | 7.0786 | 8.6564 |
| Scaled $r=10^2$ | **3.2777** | **3.2777** | **3.2777** | **3.2777** | 3.2794 | 3.4713 | 15.6909 | 21.5531 | 24.7776 | 21.1475 |
| Scaled $r=10^3$ | | **3.2777** | 3.2778 | 3.2793 | 3.4894 | 11.4180 | 21.1974 | 23.1106 | 24.4113 | 21.1934 |
| Scaled $r=10^4$ | | **3.2777** | 3.2803 | 3.4918 | 12.8537 | 18.3714 | 21.7953 | 23.3156 | 24.3576 | 21.1724 |
| Scaled $r=10^5$ | | 3.2787 | 3.5227 | 10.6500 | 18.5895 | 19.4165 | 21.8154 | 23.3321 | 24.3471 | 21.1932 |

Table 2: Averaged test loss under perturbations with standard deviation $\sigma$ across different LoRA factorizations of the same model. Lower is better.

## 11 Additional Experiment: GPT-2 fine-tuned on Wikitext2

To complement our experiments, here we provide another setting, and we report the loss under perturbations for the GPT-2 model fine-tuned on Wikitext2. These are summarized in the following table. The setting is exactly the same as what we explained for the experiments in the main body of the paper.

## 12 LLM Usage Disclosure

We used *ChatGPT 5* only for minor copyediting (grammar, wording, and clarity) during manuscript preparation. No technical content, proofs, analyses, or results were generated by the model; all ideas and conclusions are our own.

