# OpenReview forum: "Symmetries in Weight Space Learning: To Retain or Remove?"
_ICLR.cc/2026/Conference — Submitted to ICLR 2026_

### Official Review · Reviewer_eyjU · 2025-10-27

**Soundness:** 3
**Presentation:** 2
**Contribution:** 4
**Rating:** 4
**Confidence:** 3

**Summary:**

This work focuses on symmetries in the context of weight space learning. The main research question is to determine whether, in general, quotienting out weight space symmetries makes sense, depending on the downstream task.

To answer that question, the authors propose a theoretical analysis of weight space symmetries in the context of weight space regression analysis. This analysis is performed across different differentiation orders with respect to the weights. The authors first show that when taking into account only zeroth-order features, weight space symmetries hold and quotienting them out makes sense. They then show, however, that this finding does not hold for higher order features, the first example of which is the Hessian Trace, a second-order feature. They show that this finding generalises to cases where symmetries do not hold, as well as arbitrary-order features.

Finally, the authors test their findings in a small experiment on LoRA adapters for LLMs. They show that the sensitivity of the weights, a second-order feature, is not preserved under transformations of the weights that preserve the function represented by the LLM.

The conclusion of the paper is that depending on the target variable, quotienting out weight space symmetries does not make sense in all cases and can jeopardise the universality of the weight space learning model.

**Strengths:**

The authors propose a solid, complete theoretical analysis of symmetries in the context of discriminative downstream tasks in weight space learning. I particularly liked the organisation of the paper, starting from the zeroth-order case, then giving the good example of the Hessian trace, to finally prove generalisability to arbitrary-order cases.

* S1: I think the formalisation of the problem is good, and the different analyses look solid. I think they back the main point of the paper well. I appreciate the formulation of the problem using \$g \in G\$ which is very general and can adapt to most kinds of symmetries.

* S2: I am convinced that the research question, as well as the main conclusion of the paper that follows, are very timely, interesting and impactful. By knowing in which cases quotienting out weight space symmetries works, weight space learning practitioners will be able to use the right meta-models for their downstream tasks.

* S3: I like that the authors used the example of LoRAs throughout the paper, given the large number of existing equivalent LoRA matrices. It is also nicely linked with the experiment section.

**Weaknesses:**

While I remain convinced that this paper is generally strong and impactful, it has several weaknesses that justify my rating.

## Major

* W1: The experiment in Section 4 demonstrates practically that there exists some cases where even when the mapping represented by the neural network’s weights is unchanged, other characteristics such as the sensitivity can change. The existence of such cases does justify using weight space learning predictors that are NOT equivariant to such changes. The experiment setup falls short, however, of actually comparing prediction results using different weight space learning models. As such, while the experiment is useful in validating intermediate results, it is not sufficient to back the main claim of the paper. My opinion is that the authors should compare different weight space meta-model architectures to validate empirically their main claim on lines 59-62. For example, the authors could compare the performance of equivariant vs. non-equivariant meta-models on downstream tasks (such as  accuracy prediction or sensitivity analysis) to empirically validate their theoretical claims.

* W2: From my understanding, I think the definition of zeroth-order weight-space learning (Eq. 1) is limited to situations where feature extraction is done by probing the model studied with probes sampled from \$\mathcal{X}\$. Such a method is indeed, by design, invariant to any changes in the model weights that does not result in a change in the functions mapped. Many weight space learning methods, however, do not rely on probes, but rather only on the weights. For example, [1] uses weights statistics, [2] encodes weights such that they are equivariant under permutation symmetries, and [3] uses tokenised weights in a way that is not directly invariant to such symmetries. Given the importance of such methods in the field, I think this problem definition should be covered in the paper. I am relatively confident such an analysis would go in the same direction as the other results in the paper.

* W3: I think the related work section is quite limited. Several key papers in the area of weight space learning are missing. In particular, I think that [2] and [4, 5] are all key papers in the area of equivariance to weight space symmetries. The authors could extend that Section and/or add relevant citations in the introduction to better contextualise their work within the field of weight space learning. Furthermore, statements such as the start of the paper (line 32) imply that weight space learning is predicting properties from model weights. There are other applications to weight space learning such as learning to optimise [6] or neural network weights generation [7, 8]. I recommend the authors be more specific in their phrasing to make explicit that their work focuses on downstream tasks around property prediction, not weight space learning in general.

## Minor (did not impact the rating)

* W4: The paper still contains a lot of typos, in particular in the Introduction. I advise the authors thoroughly proofread their manuscript.

* W5: Regarding Table 1, I think having the “Unperturbed” baseline as a line instead of a column is confusing, since it is simply the case where \$\sigma = 0\$. In addition, on line 428, Table 4 (which does not exist) is referenced instead of Table 1. Also, the last column is \$5 \cdot 10^{-4}\$, is there a typo or are the results not in order?

## References

[1] Unterthiner, T., Keysers, D., Gelly, S., Bousquet, O., & Tolstikhin, I. (2020). Predicting neural network accuracy from weights. arXiv preprint arXiv:2002.11448.

[2] Navon, A., Shamsian, A., Achituve, I., Fetaya, E., Chechik, G., & Maron, H. (2023, July). Equivariant architectures for learning in deep weight spaces. In International Conference on Machine Learning (pp. 25790-25816). PMLR.

[3] Schürholt, K., Mahoney, M. W., & Borth, D. (2024, July). Towards Scalable and Versatile Weight Space Learning. In International Conference on Machine Learning (pp. 43947-43966). PMLR.

[4] Kofinas, M., Knyazev, B., Zhang, Y., Chen, Y., Burghouts, G. J., Gavves, E., ... & Zhang, D. W. (2024). Graph neural networks for learning equivariant representations of neural networks. arXiv preprint arXiv:2403.12143.

[5] Lim, D., Maron, H., Law, M. T., Lorraine, J., & Lucas, J. (2023). Graph metanetworks for processing diverse neural architectures. arXiv preprint arXiv:2312.04501.

[6] Knyazev, B., Moudgil, A., Lajoie, G., Belilovsky, E., & Lacoste-Julien, S. (2024). Accelerating training with neuron interaction and nowcasting networks. arXiv preprint arXiv:2409.04434.

[7] Schürholt, K., Knyazev, B., Giró-i-Nieto, X., & Borth, D. (2022). Hyper-representations as generative models: Sampling unseen neural network weights. Advances in Neural Information Processing Systems, 35, 27906-27920.

[8] Soro, B., Andreis, B., Lee, H., Jeong, W., Chong, S., Hutter, F., & Hwang, S. J. (2024). Diffusion-based neural network weights generation. arXiv preprint arXiv:2402.18153.

**Questions:**

* Q1: This is linked with W2: how do the authors think their framework works on weight-space learning models which only rely on model weights? Under which conditions could and should such models handle weight-space symmetries?

---

> ### Author Response · Authors · 2025-12-04
>
> Thanks for your review. Here is our response to the comments:
>
>
>
>
>
>
>
> >  W1
>
>
> **Answer:** We have proof-of-concept experiments in our paper, and the message is something else. We neither provided any new method nor claimed anything empirical. We just proved that expressive power is compromised under invariant learning.
>
> > W2
>
> **Answer:** The study of zeroth-order features is just a theoretical proof of concept, and the main message is beyond this. We identified an issue that using invariant/equivariant features for weight space learning under symmetries compromises the expressive power. This is already quite a strong message, and solving the difficult problem and when and how consider invariant learning is out of scope of this work. We believe such a question is too deep to be answered in just one paper, and a series of papers is required to identify and study such problems. Our message is something else; we identified an important issue and reported it in our paper as the main contribution, which, in our opinion, is well-suited for an ICLR submission.
>
>
> >  W3
>
>
> **Answer:** Thanks for the reference. If our paper gets in, we will definitely include all the suggested references in our paper to have a comprehensive literature review. Thanks!

---

### Official Review · Reviewer_WzJu · 2025-10-29

**Soundness:** 4
**Presentation:** 2
**Contribution:** 2
**Rating:** 4
**Confidence:** 2

**Summary:**

**Note:** While I am familiar with weight-space learning, and have gone over the entire paper, I have limited theoretical background and can not fully evaluate the theoretical contributions of the paper. My review therefore focuses mostly on the motivation of the paper, the practical implications of it, and empirical aspects.

The paper provides a theoretical examination of symmetries in weight-space learning tasks. The authors show that zeroth-order features (based on model outputs) can be handled with standard symmetry-aware methods. They then provide an example of sensitivity prediction on LoRAs where only a subset of the model's symmetries are relevant for learning, followed by an example where no symmetry can be preserved, concluding that any compression of the weight space can harm some downstream tasks. Lastly, they demonstrate that models which are functionally identical can have different sensitivities, validating that symmetry removal must be task-dependent.

**Strengths:**

- The paper addresses a timely question about how to handle symmetries in weight-space learning.

- The empirical evaluation is interesting. The authors demonstrate that two networks which behave the same can have very different sensitivities to perturbations. This observation could be valuable for future research on weight-space learning research as well as for building more robust networks.

**Weaknesses:**

- **Limited practical guidance.** My main concern is that the paper does not provide general guidelines for which symmetries to account for each downstream task. While the authors analyze the relevant symmetries for zeroth-order and sensitivity features, they do not offer guidance for practitioners working on other tasks. The paper would be strengthen from proposing a systematic method for identifying the relevant symmetries to account for given a specific task.

- **Narrow empirical scope.** The experimental section is very limited where only architecture and dataset are evaluated. The conclusio would be much strengthened by testing the same hypothesis on a few more cases.

- **Sectoin 3.4 accessibility.** Section 3.4 introduces many notations making it difficult to follow for readers without a strong theoretical background. This section could be clarified with more intuitive explanations to along the mathematical derivation.

**Questions:**

I dont have any further questions.

---

> ### Author Response · Authors · 2025-12-04
>
> Thanks for your review. Here is our response to the comments:
>
>
>
>
>
>
>
> > Limited practical guidance. My main concern is that the paper does not provide general guidelines for which symmetries to account for each downstream task. While the authors analyze the relevant symmetries for zeroth-order and sensitivity features, they do not offer guidance for practitioners working on other tasks. The paper would be strengthen from proposing a systematic method for identifying the relevant symmetries to account for given a specific task.
>
>
> **Answer:** The study of zeroth-order features is just a theoretical proof of concept, and the main message is beyond this. We identified an issue that using invariant/equivariant features for weight space learning under symmetries compromises the expressive power. This is already quite a strong message, and solving the difficult problem and when and how consider invariant learning is out of scope of this work. We believe such a question is too deep to be answered in just one paper, and a series of papers is required to identify and study such problems. Our message is something else; we identified an important issue and reported it in our paper as the main contribution, which, in our opinion, is well-suited for an ICLR submission.
>
>
> > Narrow empirical scope. The experimental section is very limited where only architecture and dataset are evaluated. The conclusio would be much strengthened by testing the same hypothesis on a few more cases.
>
>
>
> **Answer:** We have proof-of-concept experiments in our paper, and the message is something else. We neither provided any new method nor claimed anything empirical. We just proved that expressive power is compromised under invariant learning.
>
> > Sectoin 3.4 accessibility. Section 3.4 introduces many notations making it difficult to follow for readers without a strong theoretical background. This section could be clarified with more intuitive explanations to along the mathematical derivation.
>
>
>
> **Answer:** Since this part of the paper is just for the proof of concept, we are not sure if that is a major issue. We will address this, but we believe that the main contribution is something else and is well explained in the paper; this comment is just a minor concern.

---

### Official Review · Reviewer_nW9d · 2025-10-31

**Soundness:** 3
**Presentation:** 2
**Contribution:** 2
**Rating:** 4
**Confidence:** 3

**Summary:**

This paper studies the role of parameter symmetries in weight-space learning — learning predictors that take model parameters as inputs. The authors distinguish zeroth-order features from higher-order, derivative-based features. Key theoretical claims: zeroth-order meta-regressors inherit the model symmetry; higher-order features are invariant only to the subgroup that fixes the coefficient tensor; many natural weight-space tasks break the model’s parameter symmetries and would be lost if one quotients them out. Experiments on LoRA-fine-tuned Llama show sensitivity to re-factorizations.

**Strengths:**

1. This paper provides a clear, rigorous characterization of when weight-space functionals are symmetric.

2. Sensitivity experiments on real LLM + LoRA show that reparameterization alters higher-order quantities, and a balance transformation is proposed to improve robustness.

**Weaknesses:**

1. The paper does not clarify how the assumption of a “spectral gap” in representations is verified or estimated in real-world networks or LoRA settings. Further discussion is needed on whether this condition generalizes to realistic models.

2. The empirical evaluation focus on a single dataset/model pair; incorporating more diverse experiments would strengthen the empirical evidence and the robustness of the claims.

**Questions:**

1. When $k$ is very large or when there are special elements such as $-I$ in the group $G$, what boundary cases will invalidate the conclusions? The text has mentioned some exceptions; could you provide examples of whether these exceptions would occur in real-world networks?

2. More illustrative examples would be beneficial. Could the authors further elaborate on which specific mathematical functions might be lost if parameter symmetries are removed indiscriminately?

---

> ### Author Response · Authors · 2025-11-17
>
> We thank the reviewer for their constructive feedback.
>
> > The paper does not clarify how the assumption of a “spectral gap” in representations is verified or estimated in real-world networks or LoRA settings. Further discussion is needed on whether this condition generalizes to realistic models.
>
> **Answer:** Thanks for your comment. Let us first briefly explain the main message of our paper. While we value your comment, the main message of the paper is well beyond what Theorem 3.9 and the spectral gap mean.
>
> Our message is as follows: **Given a learning task (either regression or classification) on weight spaces, one should not consider invariant/equivariant architectures, as this will lead to the loss of expressive power.**
>
>
> What is this important? Because intuitively, and well motivated from the literature on geometric deep learning, people consider invariant features to do learning for invariant function spaces. **This intuition fails for weight space learning, and this is the main message of the paper.** For instance, a recent work [1] proposed various architectures for invariant learning for weight spaces, and our message is: **Be cautious when quotienting out symmetries in weight space learning**. In our opinion, this is an important contribution, and it is our main goal to deliver this message to the community to build foundations and methods for understanding what kind of invariant and non-invariant features are essential in weight space learning under symmetries.
>
> Our paper includes a proof of this fact for LoRAs and other cases, as well as a class of weight space functions (the so-called zeroth-order) being totally invariant. This is the main contribution.
>
> It is worth mentioning that the experiments and the theory we provided in the paper are for the theoretical/experimental proof of our concept. We do not claim to have any new architecture or to have any message on how to pick appropriate features for weight space learning under symmetries. We ask the reviewer to consider this; it is just the beginning of this line of work, and we are contributing to its foundations, though we cannot solve all essential problems in the field in this submission.
>
> Returning to the question, we provide the following answer:
>
> (1) If the action of the group $G$ is faithful (i.e., non-trivial), then there is always a positive spectral gap, meaning that $c<1$. We will follow shortly with an updated manuscript with the proof, but it essentially follows from basic properties of group action and representation theory.
>
> Moreover, testing the spectral gap in practice requires computing the trace of the representation over all group elements $g \in G$. If we know the group (i.e., rotation, scaling, permutation), one can use analytic formulas and closed-form expressions to obtain a bound on it. Thus, while in general it needs searching over the whole group $G$ (i.e., difficult), if we have a sense of what symmetries exist in the problem (which is the case for almost all ML/DL applications), then we can leverage it and find appropriate bounds.
>
> Returning to our previous comment, we kindly ask the reviewer to please consider the main message of the paper. It is well beyond the technical details of Theorem 3.9. The theory and experimental section of the paper is just a proof of the concept of a bigger message.
>
> [1] Putterman, Theo, et al. "Learning on loras: Gl-equivariant processing of low-rank weight spaces for large finetuned models." arXiv preprint arXiv:2410.04207 (2024).
>
> > More illustrative examples would be beneficial. Could the authors further elaborate on which specific mathematical functions might be lost if parameter symmetries are removed indiscriminately?
>
> **Answer:** We thank the reviewer for their comment. That is correct, a more diversified set of experiments would help, but honestly, we do not think it will change the message of the paper. We are not introducing any new architecture of solution, and we do not even claim that symmetries should be quotiented out or not. We just claim that expressive power is sensitive to such quotient space projects, and one needs to be cautious about it. Unfortunately, the deadline is now quite tight for us, and we have trouble updating the paper in a week or so with an extensive set of new experiments. We kindly ask the reviewer to consider this. This cannot change the message of the paper whatsoever.
>
> For the lost function, there is a canonical example: In Equation (12) in the paper (and the discussion after that), we proved that the gradient with respect to parameters (i.e., the so-called sensitivity with respect to parameters) is a function that is lost if we remove symmetries in the LoRA formulation. This is a concrete example. We are happy to provide more explanation or other examples here as well. Please let us know about this.

---

> ### Author Response · Authors · 2025-11-17
>
> > When k is very large or when there are special elements such as -I in the group G, what boundary cases will invalidate the conclusions? The text has mentioned some exceptions; could you provide examples of whether these exceptions would occur in real-world networks?
>
>
> **Answer:** Thanks for your comment. We guess you mean Theorem 3.8. If that is the case, then having $-I$ will not invalidate any conclusion. The only edge case is the two-element group $G = \\{I, -I\\}$, and $k$ is even. So essentially if the group has more than two elements, then there is no edge case, and the theorem is fully valid. Beyond the trivial group with two elements, all cases satisfy that. Even if we consider the two-element group, as long as $k$ is odd, the theorem still holds! The only edge case is when $G = \\{I, -I\\}$ and $k$ is even, and this is a very specific kind of group and function.  We already clearly mentioned this assumption in the paper. We can confidently say that in almost any practical setting, $G$ has at least three elements, so the assumption, in our opinion, is a very weak kind of requirement, if not trivial.
>
>
>
> Please let us know what we should do to convince you that our paper deserves acceptance. We kindly asked you to consider the main message of the paper and the importance of the fact that one needs to be cautious about removing symmetries in weight space learning, as opposed to ordinary learning with invariances.

---

> ### Author Response · Authors · 2025-11-23
> **The New Version**
>
> Dear Reviewer nW9d,
>
> We uploaded a new version of the paper, and in Remark 10.2 (newly added), we explained your concern about Theorem 3.9, along with adding pointers in the main body of the paper (Page 7 footnote).
>
> Can you let us know if you are satisfied with the answers we provided to your concerns? We provided a point-by-point response to your comments, and we honestly feel the main contribution of the paper is well beyond the spectral gap, edge cases of the theorem (which we showed in our response that are very much restricted to unrealistic settings), and experiments (since we do not have any particular message about the use of architectures and models).
>
> Our message is beyond that. We do not claim that zeroth-order features, or tensor product features, are the most important and realistic features in this setting. We only provided proof of concept theory and experiments for a very important phenomenon:
>
> **Symmetries, when it comes to the weight space learning, must not be quotiented out. If one does this, then the expressive power is (provably) compromised**
>
> We hope you value our contribution. If so, we would appreciate it if you could update your rating. Please let us know about the current state of your opinion about the paper. We understand that it is hard to return a reject score to an accept, and we hope we can successfully change your initial evaluation. Thank you in advance.
>
> Best,
> The authors

---

### Meta-Review · Area_Chair_xypu · 2025-12-24

**Summary:**

The paper considers the problem of predicting properties of neural network models directly from their weights, focusing specifically on parameter symmetries induced by group actions that leave the input--output function invariant. The central question is whether such symmetries should be removed when performing weight-space learning. The authors observe that higher-order derivatives, particularly the (symmetric) covariant $k$-tensors $D^k f(p)$, generally preserve only a very small subgroup of these symmetries---generically only the identity (under the paper’s notion of “invariance”, see remark (*) below). This observation is illustrated empirically through experiments on Low-Rank Adaptation (LoRA).


Reviewers acknowledged the timeliness of the topic, but beyond issues of presentation clarity, they consistently pointed out the limited and fragmented empirical evidence. This is not merely a matter of quantity or exposition, but of substance: the empirical section does not directly address the central question the paper sets out to resolve (nor does the theoretical part), namely whether and how symmetry removal impacts weight-space learning performance. The paper does not precisely define what is meant by “removing” or “quotienting out” symmetries, despite the fact that multiple, radically different interpretations exist, nor does it clearly specify and demonstrate in what sense performance or “expressivity” is affected.
As a result, the goal of the paper is not articulated in a sufficiently well-defined manner. From a theoretical standpoint, the results largely amount to a restatement of elementary facts from group representation theory: invariance of higher-order derivatives is only guaranteed with respect to the corresponding isotropy subgroups, which are generically minimal. Interpreted this way, the paper does not introduce new theoretical insights, but rather reiterates well-known representation-theoretic results in the context of weight-space learning. Overall, while the paper raises a timely concern, the current theoretical and empirical analyses do not yet fully connect to the problem as formulated, leaving the contribution to weight-space learning incomplete in its present form.

(*) NB: when the group action on $\mathrm{Sym}^k(\mathcal W)$ is defined as dictated by representation theory, the tensors $D^k f$ are invariant.

**Reviewer Concerns:**

The rebuttal partially addressed secondary concerns, such as clarifying specific technical points and acknowledging missing related work that could be added.

**Reviewer Scores:**

Based on the tone and content of the reviews and the subsequent rebuttal, it is likely that reviewers would not have substantially changed their scores. While several points were clarified, the core concerns, particularly regarding the limited empirical evidence supporting the main message of the paper, would likely remain.

---

> ### Public Comment · ~Behrooz_Tahmasebi1 · 2026-03-05
> **Clarification Regarding the Decision**
>
> We respectfully disagree with several aspects of the final assessment and would like to clarify the scope and contributions of the paper for readers who may read this discussion in the future.
>
> First, the decision refers to limited empirical evidence supporting the paper’s claims. We would like to clarify that the paper presents both (1) a theoretical proof-of-concept analysis and (2)  proof-of-concept experiments that directly illustrate the phenomenon studied. The purpose of the empirical section was not to provide a large-scale benchmark evaluation but rather to demonstrate the idea. From our perspective, the experiments fulfill this role.
>
> Second, the AC's summary suggests that the paper aims to answer the question of whether and how symmetry removal affects weight-space learning performance. This interpretation does not accurately reflect the goal of the work. The central message of the paper is the following:
>
> **Given a learning task defined on weight space (e.g., regression or classification over model parameters), enforcing invariance or equivariance by quotienting out symmetries can reduce expressive power and may therefore be undesirable.**
>
> The contribution of the paper is to identify and formalize this issue, which, despite the increasing attention to symmetry in machine learning, has received limited discussion in the literature. The goal of the work is therefore diagnostic rather than prescriptive: it highlights a potential drawback of equivariant models for weight-space learning.
>
> Determining when equivariant or invariant architectures should or should not be used depends strongly on the specific task and parameterization. Addressing this question in full generality would require task-specific analyses beyond the scope of a single paper. Our contribution is to demonstrate the existence and mechanism of the issue, rather than to provide a universal solution for all weight-space learning problems.
>
> We also note that the notion of “removing” or “quotienting out” symmetries was formally defined in the paper, and the impact on expressivity was analyzed both theoretically and empirically. These definitions and results were included precisely to make the discussion concrete.
>
> Finally, the decision comments suggest that the theoretical results largely restate elementary representation-theoretic facts. We would like to emphasize that the novelty of the paper lies not in advancing representation theory itself, but in applying these concepts to analyze a previously unrecognized issue in weight-space learning. This type of conceptual transfer, from established mathematical tools to new machine learning questions, is common in geometric machine learning and related areas.
>
> We hope these clarifications help future readers interpret the reviews and decisions in the appropriate context. We appreciate the opportunity to make these points for the record.

---

### Decision · Program_Chairs · 2026-01-26

Reject